# Chronic Obstructive Pulmonary Disease and Depression—The Vicious Mental Cycle

**DOI:** 10.3390/healthcare13141699

**Published:** 2025-07-15

**Authors:** Alexandru Corlateanu, Serghei Covantsev, Olga Iasabash, Liliana Lupu, Mihaela Avadanii, Nikos Siafakas

**Affiliations:** 1Discipline of Pneumology and Allergology, Nicolae Testemițanu State University of Medicine and Pharmacy, MD-2004 Chisinau, Moldova; olga.iasabash@yahoo.com (O.I.); liliana.turcan1997@gmail.com (L.L.); mihaelaavadanii09@gmail.com (M.A.); 2Department of Surgical Oncology, Federal State Autonomous Institution National Medical Research Centre “Treatment and Rehabilitation Centre” of the Ministry of Health of the Russian Federation, 115478 Moscow, Russia; kovantsev.s.d@gmail.com; 3Department of Computer Science, University of Crete, 70013 Heraklion, Greece; siafakan@uoc.gr

**Keywords:** depression, COPD, smoking, health-related quality of life, inflammation, oxidative stress

## Abstract

**Background:** Chronic obstructive pulmonary disease (COPD) is a heterogeneous lung condition marked by persistent airflow limitation and is currently the fourth leading cause of death worldwide, accounting for 3.5 million deaths in 2021. While its physical manifestations such as dyspnea, chronic cough, and sputum production are well known, its psychological impact, particularly the high prevalence of depression among patients, remains under-recognized. **Objectives:** This narrative review aims to summarize the existing data on the association between COPD and depression, analyze their pathophysiological connections, explore treatment possibilities, and highlight the interrelationships between these conditions. **Methods:** A non-systematic literature search was conducted using PubMed, Scopus, and Google Scholar. Studies, reviews, and key publications addressing the relationship between COPD and depression were selected based on clinical relevance. Findings were synthesized thematically to provide a comprehensive and critical overview. **Results:** Depression in patients with COPD is linked to worse quality of life, increased functional impairment, higher suicide risk, and poorer adherence to treatment. Contributing mechanisms include chronic systemic inflammation, hypoxemia, oxidative stress, and psychosocial risk factors such as low educational level, socioeconomic disadvantage, and comorbidities. Despite evidence of this strong association, treatment strategies remain limited and underutilized, and no unified approach has yet been established. **Conclusions:** Depression represents a major comorbidity in COPD that exacerbates both disease burden and patient suffering. Further research is essential to clarify the underlying mechanisms and to develop integrated therapeutic approaches. Enhancing our understanding and management of this comorbidity holds promise for significantly improving patient outcomes and overall quality of life.

## 1. Introduction

COPD is considered one of the primary causes of morbidity and mortality worldwide and, at the same time, has created a growing social and economic burden [1]. The prevalence, morbidity, and mortality of COPD may vary between countries depending on the risk factors populations are exposed to [2]. In 2019, there were 392 million cases of COPD globally [3]. According to the World Health Organization (WHO), COPD is the fourth leading cause of death worldwide, causing 3.5 million deaths in 2021, which represent approximately 5% of all global deaths. In addition, according to disability-adjusted life years, COPD is the eighth leading cause of poor health worldwide [4].

Smoking represents a major risk factor for COPD. According to the WHO, smoking—affecting 1.3 billion people globally—causes around 8 million deaths annually [5] and accounts for approximately 70% of COPD cases in high-income countries [6]. E-cigarette use, especially among adolescents and young adults [7], has been linked to increased respiratory symptoms, a higher risk of airway disease, and declining lung function [8]. Early studies also link e-cigarette use to lung injury [9], and Osei et al. reported a 75% higher risk of developing COPD among current users compared to never-users [10]. It is estimated that 80% of COPD patients are likely to have at least one comorbidity [11]. The relationship between COPD and mental health has recently gained significant research interest due to its impact on quality of life [12]. COPD significantly impacts mental health, with patients experiencing higher rates of depression compared to the general population. According to Atlantis et al., the relative risk of developing depression among COPD patients is 1.69 times higher than in the general population [13]. The prevalence of depression among patients with stable COPD in primary care settings varies significantly, with a range of 10% to 57% [14]. Among patients with severe COPD (FEV1 < 50% predicted), the prevalence of depression was 25.0%, compared to 17.5% in controls and 19.6% in persons with mild to moderate COPD [15].

Quality of life in COPD is significantly impacted by the progressive character of the disease. Moreover, depression in subjects with COPD worsens this impact and has been found to have the strongest correlation with self-reported health status and reduced HRQoL [16].

COPD is a chronic condition that typically requires the ongoing use of pharmacotherapy [17,18]. Nonadherence to treatment is a significant issue in COPD, with adherence rates estimated to be below 50% [19]. A meta-analysis has revealed that patients with symptoms of depression are three times more likely to be non-adherent to their prescribed medications [20]. Nonadherence to COPD treatments results in higher hospitalization rates, increased costs, and more frequent emergency department visits [21,22].

Exacerbations of COPD represent a significant challenge for worldwide healthcare systems; they are a major cause of morbidity, mortality, and reduced health status [23]. A recent study identified depression as an independent factor for acute exacerbations of COPD (AECOPD), associated with a higher risk of readmission for AECOPD (OR 2.06, 95% CI 1.28–3.31), regardless of lung function or previous severe exacerbations in the prior year [24].

## 2. Materials and Methods

This narrative review was conducted to explore the complex interplay between chronic obstructive pulmonary disease (COPD) and depression, focusing on pathophysiological mechanisms, clinical manifestations, and pharmacological implications.

### 2.1. Literature Search Strategy

A non-systematic literature search was carried out using the databases PubMed, Scopus, and Google Scholar. The following keywords and their combinations were used: „COPD” „chronic obstructive pulmonary disease”, „depression”, „anxiety”, „psychological comorbidity”, „inflammation”, „oxidative stress”, „hypoxia”, „screening tools”, „psychopharmacology”, „antidepressants”, „smoking”, „quality of life”, „gender differences”, and „elderly patients”.

### 2.2. Inclusion Criteria

Articles published between 2000 and 2024.Peer-reviewed original research, systematic reviews, meta-analyses, narrative reviews, and clinical guidelines.Studies addressing the following:
-The relationship between COPD and depression;-Shared pathophysiological mechanisms;-Screening tools for depression in COPD;-Psychiatric impact of COPD pharmacotherapy;-Sex-specific differences in comorbidity expression.


### 2.3. Exclusion Criteria

Abstracts, commentaries, and conference proceedings without full text;Animal studies and in vitro experimental studies unrelated to clinical implications;Studies focusing solely on asthma or other respiratory diseases without clear differentiation from COPD.

### 2.4. Data Synthesis and Analysis

The selected studies were thematically analyzed and grouped into the following categories:Epidemiological link between COPD and depression;Shared pathophysiological mechanisms (inflammation, hypoxia, oxidative stress);Influence of smoking and comorbidities;Screening instruments used in clinical practice;Pharmacological interactions and treatment challenges;Gender- and age-specific variations.

The data were synthesized to highlight current understanding, identify knowledge gaps, and provide guidance for future research and clinical management. The article selection process used in this review is summarized in the flowchart presented in Figure 1.

## 3. Risk Factors

Several studies have identified risk factors for depression in patients with COPD including smoking, severe dyspnea, poor HRQoL, lower education levels and socio-economic status, and an association with non-psychological comorbidities [25,26].

### 3.1. Smoking

Psychiatrists have observed that cigarette smoke seems to have an antidepressant effect [27], potentially providing temporary relief for symptoms of depression. This may explain why studies consistently report that patients with COPD who also suffer from depression or anxiety are significantly more likely to be smokers [15,28]. In the UK, 34% of people with depression, 29% of those with anxiety, and 44% of individuals with severe mental illnesses such as schizophrenia smoke tobacco [29]. The relationship between smoking and depression is, however, bidirectional. Young adults with a history of depression are more likely to start smoking compared with healthy counterparts who have no symptoms of depression. On the other hand, long-term cigarette use has been proven to heighten susceptibility to developing depression due to nicotine-induced changes in neurophysiology, including disruptions in neurotransmitter systems essential for mood regulation [30]. In addition, the prevalence of depression and anxiety tends to rise with the increasing duration of smoking (in years) and the number of cigarette pack-years [31].

Depression as a comorbidity has been shown to have a statistically significant relationship with patients’ HRQoL and smoking status [32].

### 3.2. Pulmonary Function

Spirometry is widely regarded as the gold standard for assessing lung function in patients with chronic lung diseases. Several studies have proven a significant association between reduced lung function and symptoms of depression in subjects with respiratory disease [33]. For instance, in a retrospective cross-sectional study by Park et al. [34], parameters such as FEV1 (2.76 vs. 3.01 L; *p* < 0.001), FVC (3.53 vs. 3.80 L; *p* < 0.001), and the FEV1/FVC ratio (78.8% vs. 79.5%; *p* = 0.022) were markedly lower in the group of subjects diagnosed with depression compared with subjects with no symptoms of depression. Moderate-to-severe symptoms of depression have actually been proven to have a significant and independent correlation with the progression of lung function decline in young adults [35].

### 3.3. Socio-Economic Factors

Several studies suggest that younger persons with COPD may be more susceptible to depression [25,36,37], possibly as a result of early onset of the disease, which affects their work productivity, social engagement, and quality of life. However, other research presents the opposite, that older adults are more vulnerable to depression, possibly due to the cumulative effects of chronic illness and age-related comorbidities [26,38]. A recent study showed that people with higher educational qualifications had a 40% lower risk of experiencing depression than those with a lower educational background. Additionally, the study reported younger age and a psychological history as significant risk factors for depression in COPD patients [24]. It is important to recognize that studies exploring the relationship between depression and COPD vary widely, influenced by factors such as sample size, geographic location, and cultural differences [39].

## 4. Pathogenesis and Pathophysiology

The relationship between COPD and depression can be bidirectional [13], meaning that depression can aggravate the condition of patients suffering from COPD, leading to an increased risk of exacerbation and can indirectly lead to death. The pathophysiologic mechanisms for the development of depression in patients with COPD are complex and include the following: the anxiogenic effects of hyperventilation [40,41,42], misinterpretation of respiratory symptoms [40,43,44,45,46,47], neurobiological sensitivity to CO2, lactate, and other signals of suffocation [40,43,48,49], smoking [50,51,52,53], hypoxia [54,55,56], and inflammation [57,58,59,60,61,62].

### 4.1. The Anxiogenic Effects of Hyperventilation: Misinterpretation of Respiratory Symptoms

Anxiety in COPD patients manifests as dyspnea, sweating, and tachycardia; these symptoms are often linked to fear of dyspnea attacks and death [40,41]. A complex relationship exists between dyspnea, hyperventilation, and anxiety [40], as anxiety increases the respiratory rate, worsening dyspnea through shallow breathing [43]. During hyperventilation, the arterial partial pressure of carbon dioxide (pCO_2_) falls below 30 mmHg and pH rises above 7.4, leading to cerebral vasoconstriction and reduced cerebral blood flow. These changes can result in disturbed conscious awareness, neuromuscular irritability, electrocardiographic changes, tachycardia, and frequent arrhythmias [63]. COPD exacerbates this with increased ventilatory load, reduced capacity, and neural respiratory drive [44]. When perceived respiratory effort exceeds a threshold, it triggers emotional reactions, leading to avoidance behaviors that may temporarily reduce anxiety [45,46], but can activate a ‘dyspnea-anxiety-dyspnea cycle,’ exacerbating breathlessness and impairing quality of life [47].

### 4.2. Neurobiological Sensitivity to CO_2_, Lactate, and Other Suffocation Signals in COPD and Depression

Hyperventilation that exceeds the metabolic demands of the body leads to a reduction in CO_2_ levels and, as a consequence, induces respiratory alkalosis. This, in turn, induces vasoconstriction and characteristic symptoms of panic, including feelings of numbness, breathlessness, dizziness, and tingling sensations—symptoms that can occur in healthy persons [43]. In patients with COPD, an increased respiratory rate contributes to dynamic hyperinflation. This hyperinflation, in turn, increases elastic load and the work of breathing and decreases inspiratory reserve capacity, thereby exacerbating dyspnea [48]. In severe cases of COPD, chronic hypoventilation leads to hypercapnia [43]. The resulting elevation in pCO_2_ levels stimulates the medullary chemoreceptors, which excites noradrenergic neurons and precipitates a panic response [49]. Furthermore, hypoxia is associated with the generation of lactic acid, which is strongly implicated in the provocation of panic attacks. In addition, patients with COPD and comorbid anxiety are more sensitive to both hyperventilation and lactic acid buildup [40].

### 4.3. The Role of Nicotine Dependence and Smoking in COPD and Depression

The mechanisms underlying the association between COPD and anxiety remain unclear. COPD may lead to anxiety and depression due to persistent discomfort, functional limitations, and the psychological burden of living with a chronic and often irreversible illness. Conversely, anxiety and mood disorders may contribute to COPD development—either directly or through behaviors such as smoking [64].

Adolescents with a genetic predisposition for depression or with a history of depressive episodes have an increased risk of developing COPD through nicotine addiction [65,66]. This creates a vicious cycle; depression leads to smoking, and smoking, as a major risk factor, contributes to the development of COPD. Over 70% of cigarettes are consumed by adults with at least one mental disorder [67]. Nicotine’s impact on mental health—including depression, anxiety, stress, and alcohol dependence—is mediated by the stimulation of nicotinic cholinergic receptors and subsequent release of neurotransmitters like dopamine, which reinforces dependence by producing pleasurable effects [68,69,70,71,72,73]. According to the self-medication hypothesis, individuals with psychiatric conditions may use nicotine to alleviate neurocognitive deficits and emotional symptoms [68,69]. However, chronic nicotine use can worsen or even trigger mental health disorders [74]. Importantly, smoking cessation has been shown to significantly reduce anxiety, depression, and stress compared to continued smoking [69]. The perceived calming effect of smoking is largely due to the temporary relief of nicotine withdrawal, with mood worsening between cigarettes [72].

In turn, COPD exacerbates depression through various pathophysiological mechanisms [53].

### 4.4. The Impact of Hypoxia on COPD and Depression

Chronic, subclinical hypoxemia is a common finding in patients with COPD. Low arterial oxygen saturation has been linked to the presence of periventricular white matter lesions [75], which are similarly observed in older adults with depression [76]. Several studies have also indicated a relation between neuropsychologic impairment and persistent hypoxemia, with manifestations that include both cognitive impairments and depressive symptoms [54,55]. Hypoxia appears to contribute to these effects through multiple biological mechanisms.

The body requires nearly optimal blood oxygen levels to synthesize serotonin from its amino acid precursor, tryptophan [77]. Even slight reductions in oxygen or tryptophan availability can acutely decrease serotonin levels [78], which may contribute to the emergence of depression, impulsivity, and self-injurious behavior [79]. Furthermore, hypoxia may globally suppress cerebral energy metabolism, further promoting depressive states [80].

Research on sleep apnea has provided key insights into the relationship between hypoxemia and depression, suggesting that intermittent nocturnal hypoxemia is considered a significant cause for developing depressive states [56].

### 4.5. The Role of Inflammation in COPD and Depression

Various studies have found that pro-inflammatory cytokines, such as IL-6, affect the brain, contributing to depressive symptoms in COPD patients [56]. Elevated IL-6 levels are found in both COPD and depression, with IL-6, IFN-γ, and IL-2 involved in producing symptoms of depression [57].

IL-6 is produced at sites of inflammation and plays a key role in the acute phase response. It amplifies chronic inflammation by stimulating T- and B-cells [58]; therefore, high concentrations of IL-6 suggest a relation between depression and COPD. Additionally, IL-6 modulates monocyte and macrophage differentiation [59], which is a key component of the immune system. When activated by pro-inflammatory signals such as IFN-γ or lipopolysaccharides, macrophages release nitric oxide (NO) [59]. Activated macrophages, in addition to producing NO, also secrete neopterin, a molecule that serves as a biomarker for T helper cell activation [60]. Elevated neopterin levels have been consistently identified in both COPD patients [58] and in subjects with depression [61], and thus underline the interconnected nature of chronic inflammation, COPD, and mood disorders. In humans, the administration of IFN-γ has been shown to induce various depression-like symptoms, such as headache, weight loss, fatigue, anorexia, irritability, and difficulties in concentration [81]. Similarly, IL-2 can cross the blood–brain barrier, causing cognitive and motor impairments [62].

Taken together, these findings highlight the significant role of inflammation, particularly mediated by cytokines like IL-2, IL-6, and IFN-γ, in the pathogenesis of both COPD and depression.

### 4.6. Oxidative Stress

Oxidative stress, resulting from an imbalance between reactive oxygen species (ROS), reactive nitrogen species (RNS), and antioxidant defenses, plays a key role in the pathogenesis of COPD. In the lungs, oxidants damage nucleic acids, lipids, and proteins, trigger redox-cycling reactions, deplete antioxidants (e.g., glutathione), initiate carcinogenesis, and inactivate protease inhibitors such as α1-antitrypsin [82]. Cigarette smoke is a major exogenous oxidant source and significantly contributes to COPD [83,84].

Air pollution, containing nitrogen dioxide (NO_2_), ozone, polycyclic aromatic hydrocarbons (PAHs), and endotoxins, also induces oxidative stress and is a significant cause of COPD development [85].

Endogenous oxidant sources include mitochondria, which generate superoxide, a key ROS, along with membrane-bound oxidases like cytochrome P450 [86,87]. Increased plasma lipid peroxidation markers, such as malondialdehyde, and biomarkers like exhaled ethane and 8-isoprostanes, indicate oxidative damage in COPD patients [88,89].

In conclusion, the relationship between COPD and depression is deeply bidirectional, supported by multiple interrelated pathophysiological mechanisms. On one hand, depression promotes risky behaviors such as active smoking, which is the main etiological factor for the development and progression of COPD. On the other hand, COPD exacerbates depressive symptoms through chronic hypoxia, systemic inflammation, oxidative stress, and neurobiological dysfunctions.

These factors create a self-perpetuating vicious cycle, where the worsening of one condition drives the progression of the other, with a significant impact on patients’ quality of life and clinical prognosis.

All these mechanisms and their interactions have been synthesized and illustrated in Figure 2, providing a visual overview of the COPD–depression vicious cycle.

## 5. Depression and COPD Exacerbations

The progression of COPD is largely influenced by the frequency of exacerbations and the presence of comorbidities, both of which significantly impact the disease course and worsen prognosis [90,91,92]. Depressive symptoms in COPD patients are associated with more frequent severe exacerbations, reduced physical activity, increased dyspnea, and a lower quality of life, suggesting that depression may contribute to faster disease progression [93].

A recent study found a high prevalence of depression in COPD patients with frequent exacerbations, with more severe depressive symptoms observed in advanced disease stages [94]. The pathophysiological mechanisms involved in the impact of depression on acute AECOPD remain poorly understood [93]. Psychophysiological, behavioral, and psychosocial factors likely contribute to this association. Depression, marked by hopelessness and fear, can reduce self-care, treatment adherence, and increase smoking [95,96] all of which may contribute to AECOPD. Cognitive impairments may also amplify dyspnea perception, raising healthcare utilization and hospitalization risk [93].

## 6. Depression and COPD in the Elderly

Depression is a common problem for older people. According to reports, 80% of older adults with COPD suffer from depression [97,98]. Older adults with COPD are more likely to experience dyspnea after physical activity, leading to a long-term decline in activity levels, which in turn results in reduced muscle mass and function [99].

Patients with both COPD and comorbid depression tend to have a poorer prognosis, including lower exercise tolerance, greater functional limitations, more frequent acute exacerbations, and an increased risk of mortality compared to those without depression [100].

Depression in patients with COPD was associated with length of hospital stay and increases in both 30- and 60-day hospital readmission rates [101]. Older adults with COPD exhibit a higher prevalence of severe social disengagement (4.5% vs. 2.1%; adjusted odds ratio [OR], 0.7; 95% confidence interval [CI], 0.1–4.8) and loneliness (57.7% vs. 42.1%; unadjusted OR, 1.9; 95% CI, 1.4–2.5) compared to their counterparts without COPD [102].

## 7. COPD and Depression: Clinical Differences Between Men and Women

The global prevalence of COPD is estimated at approximately 12%, and its burden continues to rise worldwide [103,104]; furthermore, COPD has become the leading cause of death among female smokers [102]. Biological differences between males and females include differences in airway development, inflammatory responses, and susceptibility to inhaled substances such as tobacco smoke [105]. One key distinction is that females tend to have smaller airways relative to lung volume compared to males [106]. This may partially explain why women experience more severe small airway disease than men, even with comparable tobacco smoke exposure [107].

Female sex is a significant risk factor for depression in COPD, along with lower BMI, living alone, smoking, and greater disease severity, particularly in GOLD stage III/IV. Risk factors for depression differ by sex; in males, they include low BMI, low income, living alone, and multiple comorbidities, while in females, they involve lower education, urban living, and smoking [108]. Women with COPD report higher levels of depression and reduced quality of life compared to males [109], even after adjusting for lung function, age, smoking history, and emphysema severity [110]. Additionally, female sex, depression, and anxiety are linked to increased exacerbations and mortality [107].

## 8. Screening for Depression in COPD

The Global Initiative for Chronic Obstructive Lung Disease (GOLD) recommends the use of the COPD Assessment Test (CAT) for evaluating patients with COPD [111]. Research suggests that the CAT may serve as an indicator of major depression in COPD patients with mild hypoxemia [112]. Although various screening tools are available to identify depression— such as the Beck Depression Inventory (BDI) (second revision), the Center for Epidemiologic Studies Depression Scale (CES-D), the Geriatric Depression Scale (GDS), the Hospital Anxiety and Depression Scale (HADS), and the Patient Health Questionnaire-9 (PHQ-9) [113]—none of these instruments are routinely implemented in primary care practices [34,114]. Table 1 provides a comparative overview of the most commonly used scales for assessing depression in COPD patients, highlighting their structure, target population, scoring system, interpretation criteria, and limitations.

## 9. Impact of COPD Medication on Depression and Vice Versa

Managing anxiety and depression in COPD needs a multidisciplinary approach. COPD progression impacts mental health, and targeted strategies such as smoking cessation [117], influenza vaccination [118], oxygen therapy [119], pharmacological interventions [120], surgical procedures for select patients [121], structured exercise programs to condition peripheral muscles [122], and self-management strategies [123] becomes more pronounced.

While medication for COPD is essential in alleviating respiratory symptoms, some conventional therapies may lead to adverse effects. The use of corticosteroids is strongly associated with psychiatric and neurological side effects, with reported rates of depression (40.5%), mania (27.8%), psychosis (13.9%), and delirium (10.1%) [124]. A case study documented severe mania induced by the combined use of prednisone and clarithromycin [125].

Although rare, such effects require careful monitoring. Another study assessed the impact of long-acting beta-agonists (LABAs) and theophylline on suicidal ideation, finding no significant link with LABAs but an increased risk with theophylline, underscoring the need for psychological risk assessment [126]. Beta-2 adrenergic agonists, such as albuterol, indacaterol, and salmeterol, are known to cause dose-dependent prolongation of the QT interval and potassium depletion. Similarly, serotonin reuptake inhibitors (e.g., escitalopram, citalopram, fluoxetine) and tricyclic antidepressants (e.g., nortriptyline, doxepin) also have the potential to prolong the QT interval. When used together, these medications may increase the risk of ventricular arrhythmias, including torsade de pointes, and could elevate the risk of sudden death [127]. Pulmonary rehabilitation (PR) is a key COPD treatment approved by the American Thoracic Society and the European Respiratory Society [128]. It benefits patients with moderate-to-severe disease, functional limitations, and stable conditions without severe comorbidities [129]. PR improve independence, quality of life, and symptom control [130]. Beyond respiratory benefits, it also reduce depression and cognitive decline [131]. According to its effectiveness, PR should be included in treatment plans for all COPD stages [132,133].

Even with the high prevalence of depression in people with COPD and the impact on quality of life, research on its management remains limited [134]. With the pressing need for effective treatment options, it is essential to consider factors such as the risk of respiratory depression, potential side effects, and possible drug interactions when prescribing antidepressants [135].

Selective serotonin reuptake inhibitors (SSRIs) are considered the first-line treatment for depression in COPD patients. There is theoretical concern that SSRIs and SNRIs may adversely affect respiratory function in COPD through several mechanisms. Fatigue and sleepiness, experienced by 10–20% of patients, could lead to respiratory depression, hypoxemia, and hypercapnia, increasing the risk of exacerbations. Similarly, nausea and vomiting—reported by 10–30% of users may raise the likelihood of aspiration and respiratory infections [136]. Additionally, these medications may impair immune cell function, making patients more susceptible to infections [137]. Elevated extracellular serotonin caused by SSRIs and SNRIs can also reduce the clearance of apoptotic cells [138], promoting airway inflammation and obstruction, which further contributes to infections and exacerbations [139]. The initiation or use of SSRIs, serotonin norepinephrine reuptake inhibitors (SNRIs), and tricyclic antidepressants (TCAs), whether individually or in combination, has been significantly associated with an elevated risk of pneumonia. Particular attention is needed when prescribing tricyclic antidepressants (TCAs) and mirtazapine to COPD patients with hypercapnia. Current evidence suggests that the strong anticholinergic effects of tricyclic antidepressants (TCAs), which cause dry mouth, may increase the risk of pneumonia in elderly individuals [140]. Likewise, benzodiazepines can induce respiratory depression, posing a significant risk for COPD patients who retain CO_2_ [141]. Frequently, bupropion, a dopaminergic agent, is prescribed for smoking cessation. However, data suggest that dopamine agonists may impair ventilatory responses to hypoxemia and hypercapnia by inhibiting carotid-body chemoreception through dopamine-mediated mechanisms [101]. Medication tolerance should be assessed over 1–3 weeks, with psychiatric referral for suicidal behavior or severe psychiatric comorbidities [142]. To enhance clarity and facilitate clinical interpretation, the pharmacological management of COPD and its psychiatric comorbidities is summarized in Table 2.

## 10. Future Perspectives

Despite the impact of depression on COPD prognosis, clinical guidance on its management is limited. Future research should integrate mental health into COPD management, with regular screening for depression. Long-term efficacy of treatment needs to be assessed in large studies, including personalized pharmacological approaches with fewer side effects. By addressing these gaps, a more comprehensive and patient-centered approach to COPD care can be developed, ultimately improving both physical and mental health outcomes.

## 11. Conclusions

In conclusion, depression is a major yet often underrecognized comorbidity in patients with COPD, significantly exacerbating disease burden, reducing treatment adherence, and impairing quality of life. This narrative review highlights the bidirectional relationship between COPD and depression, driven by shared pathophysiological mechanisms such as systemic inflammation, hypoxia, oxidative stress, and the neurobiological consequences of smoking. Depression in COPD is associated with more frequent exacerbations, increased hospitalizations, and higher mortality rates. Sex and age differences further influence the expression and severity of depressive symptoms, underscoring the need for personalized approaches. Management is further complicated by the risk of pharmacological interactions and adverse effects of antidepressants, especially in patients with advanced disease. Despite the availability of validated screening tools, routine assessment for depression remains infrequent in clinical practice.

## Figures and Tables

**Figure 1 healthcare-13-01699-f001:**
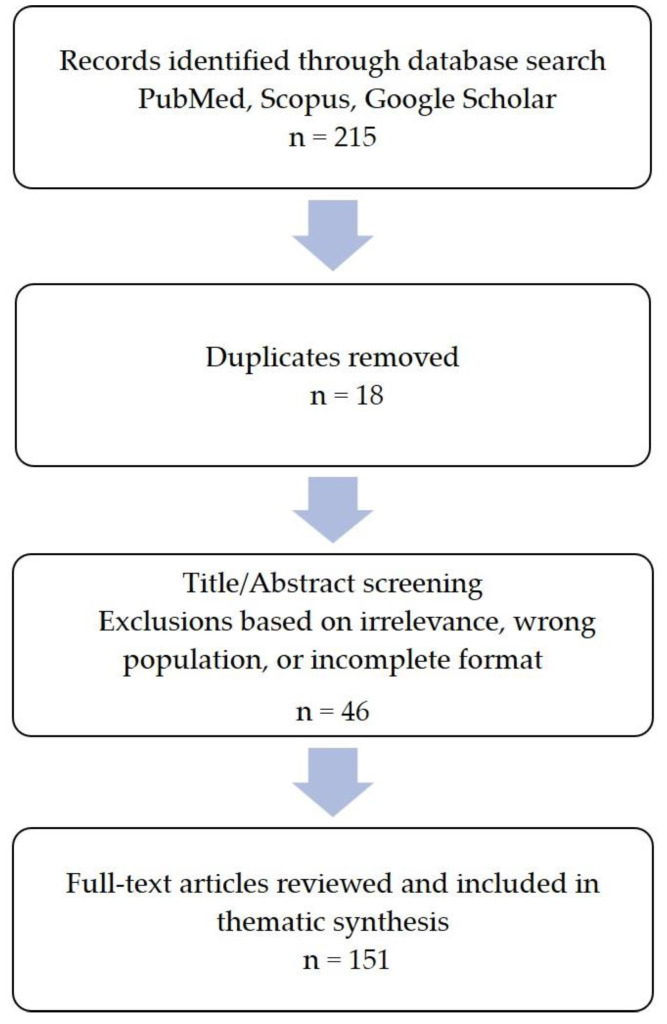
Flowchart of article selection.

**Figure 2 healthcare-13-01699-f002:**
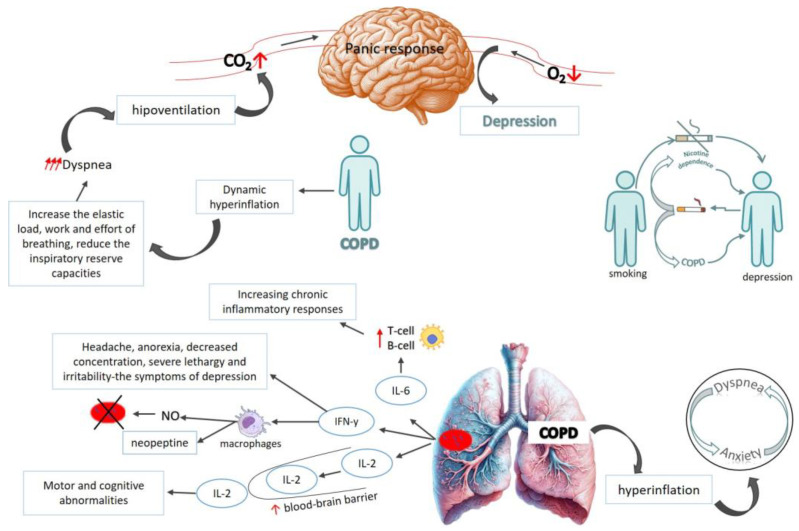
Pathogenesis and pathophysiology of depression in COPD. The bidirectional relationship between COPD and depression involves overlapping pathophysiological mechanisms; chronic inflammation (IL-6, IFN-γ, IL-2), hypoventilation, hypercapnia, and hypoxia contribute to depressive symptoms via neuroimmune and cognitive pathways. Dyspnea and dynamic hyperinflation exacerbate anxiety, while nicotine dependence links smoking, depression, and COPD in a vicious cycle. Abbreviations: T-cell—T lymphocyte; B-cell—B lymphocyte; IL-2—interleukin-2; IL-6—interleukin-6; IFN-γ—interferon-gamma; NO—nitric oxide; O_2_—carbon dioxide; O_2_—oxygen.

**Table 1 healthcare-13-01699-t001:** Scales/scores for assessment of depression in COPD.

	BDI	GDS	CES-D	HADS
**Number of questions/time for evaluation**	- 21 items - 2 past weeks	- 30 items (long version) - 15 items (short version) - current/past week	- 20 items [115] (and other versions) - 1 past week	- 14 items (7-anxiety-related, 7-depression-related)
**What is evaluated**	patients with somatic, affective, cognitive and vegetative symptoms	normal community-dwelling elderly and elders hospitalized for depression	patients with positive/negative effect, somatic problems, evaluating their activity level	psychiatric and medical patients, including cancer, traumatic brain injury, cardiac, stroke, intellectual disabilities, epilepsy, chronic obstructive pulmonary disease, etc., ages 16–65 years
**Scale**	- a 4-point scale: 0-means not at all 3-extreme form of each symptom	- a scale—“yes or no”	- a 4-point scale: 0-rarely(<1/day) 1-some/little of the time (1–2 days) 2-occasionally or a moderate amount of time (3–4 days) 3-most or all of the time (5–7 days)	- a 4-point Likert scale, ranging from 0–3
**Score interpretation**	- minimal range = 0–13-mild depression = 14–19 - moderate depression = 20–28 - severe depression = 29–63	- long form: 0–9-normal, 10–19-mild depression, 20–30-severe depression - short form: >5-suggestive for depression and >10-highly likely depression	easily hand scored. The items should be summed to obtain a total score.	- 0–7-normal - 8–10-mild - 11–15-moderate - >=16-severe
**Time to administer/complete**	- self-administration = 5–10 min. - oral administration = 15 min.	- long version—5–10 min. - short version—2–5 min.	10 min	<=5 min. (1–2 min.)
**Response format**	0–3 rating scale	«yes» or «no»	4-point Likert scale	- 0–3 rating scale
**Sensitivity to change**	- 5-point difference = minimally important clinical difference - 10–19 points = moderate difference - >20 points = large difference [116]	- the short form shows the sensitivity of 81.3% - the long form shows the sensitivity of 77.4%	- ranges of 13–21 have been provided for detecting of 80–90% reliable change.	- sensitivity = 56–100%
**Restrictions/** **limitations**	Overlapping symptoms between other medical conditions and depression, cost, and reading level	It is valid in younger samples. It needs caution when used with cognitively impaired individuals and severely cognitively impaired individuals	Response format can be difficult in the original 20-item instrument and is a contributing reason for the development of shorter versions	It is better to compare HADS to other measures of depression
**Ease of use**	time to complete—5–15 min.	self-administered questionnaire	time to interpret < 10 min. Easily self-administered/administered by interviewer	self-administered questionnaire

**Table 2 healthcare-13-01699-t002:** Pharmacological considerations in COPD and depression. This table summarizes the pharmacological interventions commonly used in COPD management. The table also includes antidepressants used in comorbid depression and their respiratory impact, emphasizing the need for careful monitoring of psychiatric and respiratory side effects.

Therapy Category	Respiratory/Psychiatric Effects
Corticosteroids	Depression, psychosis, mania, anxiety, delirium. Reported rates: depression (40.5%), mania (27.8%), psychosis (13.9%), delirium (10.1%). Severe mania with combination therapy.
Long-acting beta-agonists (LABAs)	No significant link with suicidal ideation.
Methylxanthines	Increased risk of suicidal ideation.
Beta-2 adrenergic agonists	Dose-dependent QT prolongation and potassium depletion. Combined use with antidepressants increases risk of ventricular arrhythmias and sudden death.
Selective serotonin reuptake inhibitors (SSRIs)	Fatigue, sleepiness (10–20%) leading to respiratory depression, hypoxemia, hypercapnia. Nausea, vomiting (10–30%) increasing risk of aspiration and respiratory infections. Impaired immune cell function. Elevated serotonin may reduce clearance of apoptotic cells, leading to airway inflammation and obstruction. Increased pneumonia risk.
Serotonin-norepinephrine reuptake inhibitors (SNRIs)	Similar respiratory and psychiatric effects as SSRIs. Increased pneumonia risk.
Tricyclic antidepressants (TCAs)	QT prolongation. Increased pneumonia risk, especially in hypercapnic and elderly patients due to anticholinergic effects. Risk of ventricular arrhythmias and sudden death when combined with beta-agonists or SSRIs.
Atypical antidepressants	Particular caution needed in hypercapnic patients.
Benzodiazepines	Can induce respiratory depression. High risk in patients with CO_2_ retention.
Dopaminergic agents	May impair ventilatory responses to hypoxemia and hypercapnia via dopamine-mediated inhibition of carotid-body chemoreception.
Pulmonary rehabilitation (PR)	Reduces depression and cognitive decline. Improves independence, quality of life, and symptom control.
General antidepressant therapy	Elevated pneumonia risk. Psychiatric referral advised for suicidal behavior or severe comorbidities. Risk of respiratory depression, aspiration, immune suppression, infections, and exacerbations.

## Data Availability

Data sharing is not applicable. No new data were created or analyzed in this study.

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
