# Peer review of "Chronic Obstructive Pulmonary Disease and Depression—The Vicious Mental Cycle"

_healthcare, 2025, doi:10.3390/healthcare13141699_

Round 1
Reviewer 1 Report
Comments and Suggestions for Authors
The topic addressed in this work is undoubtedly of clinical interest, and it would have been valuable to further explore the vicious cycle linking depression and COPD. However, this aspect appears to have been only briefly mentioned, while the manuscript tends to introduce several different themes without delving into them in sufficient depth.
Abbreviations should be spelled out upon first use—for example, COPD should be defined in the introduction and not fully written out again later, once the abbreviation has already been used multiple times. I would also recommend using the full term "Chronic Obstructive Pulmonary Disease" in the title.
There are a few minor typographical errors throughout the manuscript. For instance, in the paragraph on pulmonary function:
“Specifically, parameters such as FEV1 (P<0.001), FVC (P<0.001), and the FEV1/FVC ratio (P=0.022) were markedly lower in the group of subjects diagnosed with depression compared with subjects with no symptoms of depression.”
While P-values are reported, the corresponding study is not described—further clarification is needed here.
Additionally, the methodology section is lacking: it is unclear how the research was conducted, what type of review this is, and whether appropriate guidelines for reviews were followed.
It would also be important to clarify how this work differs from the numerous existing studies on the relationship between COPD and depression, as this comorbidity has already been extensively explored.
The quality of the figure is low, and it is difficult to read. Some words appear to be highlighted without explanation—this should be corrected. Regarding the table, the row labeled “are evaluated” is unclear and likely needs rephrasing.
Overall, the manuscript touches on many relevant points, but would benefit from a more focused and in-depth discussion of key mechanisms—particularly the smoking–depression–COPD vicious cycle, which deserves greater attention.
Author Response
Response to Reviewer
We would like to sincerely thank the reviewer for their thorough and constructive comments, which have been invaluable in improving the quality and clarity of our manuscript. We greatly appreciate the time and effort dedicated to providing this detailed feedback. Please find below our responses to the points raised:
- Vicious cycle linking COPD and depression:
We fully agree that this interplay represents a critical clinical and pathophysiological aspect. In the revised manuscript, we would like to emphasize that the vicious cycle was addressed in detail in the “Pathogenesis and Pathophysiology” section. Specifically, we have described the bidirectional relationship between COPD and depression, highlighting how depression can exacerbate COPD symptoms, increase the risk of exacerbations, and negatively influence prognosis. Conversely, we also discussed how COPD contributes to the onset and worsening of depression through multiple interconnected mechanisms. To provide a structured and in-depth exploration, we detailed six key pathophysiological pathways contributing to this vicious cycle. These mechanisms were supported by relevant studies and referenced accordingly, aiming to offer a holistic view of the complex COPD-depression relationship. However, in response to the reviewer’s suggestion, we have now enhanced the visibility and clarity of this analysis by: Summarizing the “vicious cycle” concept more explicitly at the end of the Pathophysiology section, to ensure the bidirectional nature of the relationship stands out more clearly. - Abbreviations and title:
We revised the manuscript to ensure all abbreviations are spelled out at first mention and updated the title to include “Chronic Obstructive Pulmonary Disease” in full. - Typographical errors and P-value clarification:
We have carefully reviewed the manuscript and corrected minor typographical errors identified by the reviewer. Regarding the paragraph on pulmonary function, we have added the necessary clarifications concerning the origin of the reported P-values and provided more detailed information about the referenced study to enhance transparency and reproducibility. - Methodology section:
We acknowledge that the methodology section lacked sufficient detail and clarity. In response, we have revised the Methods section to explicitly describe the type of review and the approach taken. Specifically, we clarify that this is a narrative literature review, focused on summarizing current evidence regarding the association between COPD and depression. - Figure and table quality:
The figure has been replaced with a higher-quality version to ensure readability, and unnecessary highlights have been removed for clarity. Additionally, the table has been revised, and the ambiguous row labeled “are evaluated” has been rephrased for better understanding.
Once again, we would like to express our gratitude for the reviewer’s constructive critique, which has greatly contributed to enhancing the overall quality of our manuscript.
Respectfully,
Corlateanu A.
On behalf of all authors
Reviewer 2 Report
Comments and Suggestions for Authors
• The authors have to elaborate on the molecular impact of smoking on both COPD and depression
• The manuscript needs to provide a global perspective of the research problem, i.e. more data have to be added about the prevalence of COPD and depression globally as this is related to smoking please add information about smoking
• Although COPD medications were added in a section, however, this should be added as table or figure to make it easier to the readers
• In addition, antidepressants were not mentioned in the manuscript, do they affect COPD status?
Author Response
Dear Reviewer,
Thank you very much for your valuable and insightful feedback.
Please find below our detailed responses to your comments:
- “The authors have to elaborate on the molecular impact of smoking on both COPD and depression.”
The manuscript already addresses this topic in detail within the section “Pathogenesis and Pathophysiology”, particularly in subsections 3.3 (nicotine dependence and smoking), 3.4 (hypoxia), and 3.5 (inflammation). These sections cover relevant molecular mechanisms such as neuroinflammation, oxidative stress, hypoxia-induced alterations, and the role of pro-inflammatory cytokines (e.g., IL-6, IFN-γ, IL-2) and oxidative stress markers — all of which contribute to both COPD and depression.
- “The manuscript needs to provide a global perspective of the research problem, i.e. more data have to be added about the prevalence of COPD and depression globally as this is related to smoking. Please add information about smoking.”
Thank you for highlighting this aspect. We have revised the Introduction section to include a more comprehensive global perspective on the prevalence of COPD and depression. In addition, we have integrated recent statistics on global smoking rates and emphasized the established link between tobacco use and both conditions. These additions help contextualize the significance of the problem from a global health standpoint and reinforce smoking as a key shared risk factor.
- “Although COPD medications were added in a section, however, this should be added as table or figure to make it easier to the readers.”
Thank you for this practical suggestion. We have addressed your comment by reorganizing the content on COPD pharmacological treatment into a dedicated table. The new table clearly summarizes the main classes of medication, examples, mechanisms of action, and relevant psychiatric side effects. We believe this format improves clarity and enhances reader engagement with the material.
- “In addition, antidepressants were not mentioned in the manuscript, do they affect COPD status?”
We would like to clarify that the impact of antidepressants on COPD has already been discussed in our manuscript. Specifically, we included a paragraph that outlines:
- The clinical challenges of treating depression in patients with COPD, including the risks of respiratory depression and drug interactions.
- The role of selective serotonin reuptake inhibitors (SSRIs) as first-line agents and their potential to inhibit cytochrome P450 enzymes.
- The association between SSRIs, SNRIs, TCAs and an increased risk of pneumonia.
- Caution regarding the use of TCAs, mirtazapine, and benzodiazepines in patients with hypercapnia.
- The utility of bupropion in smoking cessation and its influence on ventilatory response.
- The importance of individualized treatment tolerance assessments and timely psychiatric referrals.
We believe this content offers a balanced and concise overview of the topic, consistent with the scope of our narrative review.
We are grateful for your constructive comments, which have contributed to strengthening the manuscript.
With respect and appreciation,
The Authors.
Reviewer 3 Report
Comments and Suggestions for Authors
Introduction:
- Lines 31-33: Please check the use of the "." in the reference number.
- Lines 35-37: Only reference number 3 should be cited; there is no need to repeat it.
- The manuscript does not indicate whether this is a systematic or narrative review.
Methods:
- There is no clear methods section explaining how the data were collected or how the literature was selected, which is essential for validating the literature review.
- How does this manuscript specify the inclusion and exclusion criteria for the literature used?
- There is no section detailing the databases utilized for the search.
- What analysis framework is employed (e.g., PRISMA for systematic reviews)?
Results and Discussion
- The manuscript lacks a specific section that discusses the results and discussion by journal guidelines, which state: "All review papers should have the following structure: Abstract, Keywords, Introduction, Methods, Results, Discussion, and Conclusions."
- Please review the guidelines for writing review papers at https://www.mdpi.com/journal/healthcare/instructions#template.
References:
- The reference formatting is inconsistent. Please ensure that references conform to the journal's guidelines.
Author Response
Thank you for your careful review and constructive comments. We have made the following revisions to address your observations:
Introduction
- The punctuation error related to the placement of the period within the reference number in lines 31–33 has been corrected.
- The redundant citation of reference [3] in lines 35–37 has been removed to ensure clarity and avoid unnecessary repetition.
- To improve transparency regarding the manuscript type, we have explicitly specified in the Abstract that the present article is a narrative review, not a systematic one.
Methods:
We acknowledge that the methodology section lacked sufficient detail and clarity. In response, we have revised the Methods section to explicitly describe the type of review and the approach taken. Specifically, we clarify that this is a narrative literature review, focused on summarizing current evidence regarding the association between COPD and depression.
Results and Discussion
We appreciate your observation. As the current manuscript is designed as a narrative review, its structure follows the Healthcare journal’s guidelines, which allow thematic organization of content without the obligation to include a distinct “Results” section. According to the journal’s instructions for authors, such a structure is appropriate for non-systematic reviews, where the literature is synthesized within logically organized sections and subsections. Therefore, we have opted for a content-driven format that aligns with the narrative nature and scope of this review.
References:
We have carefully revised the entire reference list to ensure consistency with the journal's citation style. All references have been reformatted in accordance with the Healthcare journal’s guidelines.
Thank you again for your constructive suggestion and for engaging with our work.
With respect and appreciation,
Corlateanu A.
On behalf of all authors
Reviewer 4 Report
Comments and Suggestions for Authors
Very interesting work. It's important to follow a scientific systematic review methodology. I suggest you move forward with the PRISMA methodology.
Author Response
Thank you very much for your feedback and appreciation of our work.
We would like to clarify that this article was intentionally designed as a narrative review, aiming to provide a broad and integrative perspective on the complex relationship between COPD and depression. While we acknowledge the value of systematic methodologies such as PRISMA, our objective was to explore key themes and conceptual frameworks rather than conduct a quantitative synthesis or meta-analysis.
That said, we ensured a rigorous selection and critical evaluation of the literature, based on relevance, credibility, and recentness. We will emphasize the narrative nature of the review more clearly in the revised manuscript to avoid any confusion regarding the methodology.
Thank you again for your constructive suggestion and for engaging with our work.
Respectfully,
Corlateanu A.
On behalf of all authors
Round 2
Reviewer 1 Report
Comments and Suggestions for Authors
The abstract methods section contains typos and remains overly generic. Furthermore, the search strategy is not clearly described.
There are still abbreviation errors: for example, "COPD" is used in the introduction without being defined.
A dedicated "Materials and Methods" section is still missing. In their response to my previous comment, the authors stated that they had enriched this section; however, it was never included and is still not present in the manuscript. This is a major methodological limitation.
The figure remains scientifically inadequate: acronyms are not explained, inconsistent fonts and bolding are used, and some labels remain difficult to read.
In the new Table 2, several fields in the “Respiratory impact” column are left blank. This lack of completeness compromises the scientific value of the table and introduces ambiguity.
Moreover, the discussion of the relationship between COPD and depression remains superficial and insufficiently developed. Each paragraph only offers brief mentions rather than a thorough and in-depth analysis.
Overall, it appears that the authors have reviewed the previous comments hastily and addressed them only superficially, without implementing meaningful changes to the manuscript. The absence of a proper methods section, despite the authors’ claim to have expanded it, is the clearest evidence of this.
Author Response
Dear Reviewer,
Thank you very much for your careful reading of my manuscript and for your constructive and insightful feedback. I truly appreciate the opportunity to revise and improve the article. Please find below my detailed responses to each of your observations:
Comment 1 & 3: I fully acknowledge that the previous version of the Methods section was insufficiently detailed. In response, I have carefully revised and substantially expanded the “Materials and Methods” section. The revised version now clearly describes the review process, including:
-
the databases consulted (PubMed, Scopus, Google Scholar),
-
the specific keywords and combinations used,
-
clearly defined inclusion and exclusion criteria,
-
the thematic framework for data synthesis and analysis.
I believe these additions address your concerns regarding methodological clarity, transparency, and reproducibility. I apologize for the earlier omission and confirm that a fully structured section is now included in the manuscript.
Comment 4: I have redesigned the figure to improve both its scientific clarity and visual consistency. The updated figure now includes:
-
a full legend with all acronyms explained,
-
uniform font sizes and styles throughout,
-
clearly legible labels and consistent formatting.
Comment 5: In response, I have restructured Table 2 by merging the “Respiratory” and “Psychiatric” effects columns into a single, integrated column titled “Respiratory and Psychiatric Effects”. This adjustment eliminates the blank fields and ensures uniform coverage of all therapeutic categories, while also improving the table’s clarity and clinical relevance.
Comment 6: I appreciate your feedback regarding the depth of the discussion. I agree that the original draft did not sufficiently explore this complex association. Accordingly, I have revised the discussion section to provide a more detailed and structured analysis, which is now divided into key thematic subsections:
-
4.1. The Anxiogenic Effects of Hyperventilation. Misinterpretation of Respiratory Symptoms – discussing the physiological and emotional cycle triggered by dyspnea and anxiety.
-
4.3. The Role of Nicotine Dependence and Smoking in COPD and Depression – focusing on the neurobiological and behavioral links between nicotine use and depressive symptoms.
-
4.4. The Impact of Hypoxia on COPD and Depression – examining the effects of both chronic and intermittent hypoxemia on neurotransmitter synthesis and cerebral function.
I trust this new structure offers the more comprehensive and evidence-based analysis you requested.
Comment 7: I sincerely regret if my previous revision seemed superficial. I have now taken great care to revise the manuscript thoroughly, addressing each of your comments with the attention they deserve.
Thank you again for your valuable guidance, which has significantly strengthened the manuscript.
With respect and gratitude,
Corlăteanu A.
(On behalf of all authors)
Reviewer 2 Report
Comments and Suggestions for Authors
The authors responded to the comments
Author Response
Thank you very much for your feedback!
Reviewer 3 Report
Comments and Suggestions for Authors
Authors have made improvements according to reviewer suggestions
Author Response
Thank you very much for your feedback!
Reviewer 4 Report
Comments and Suggestions for Authors
I suggest you focus on:
1. Write a specific point explaining the narrative method in detail, including academic citations to support it. Explain in detail the article selection process and how you processed and analyzed the information. Create a flowchart.
2. A specific point about the discussion of the topic covered in relation to what has already been researched and published on the subject.
3. A specific point about the limitations of your research.
4. Clear and concise conclusions about your findings.
Author Response
Dear Reviewer,
Thank you for your constructive feedback and valuable suggestions. We carefully considered all points raised and have revised the manuscript accordingly.
Regarding the narrative method, we have now provided a detailed explanation of our methodological approach, including the criteria for article selection, data collection, and thematic analysis. A flowchart has been added to clearly illustrate the selection and review process. The conclusions have been revised to present a more concise and focused summary of the main findings, emphasizing their relevance and contribution to the field.
We appreciate your helpful input, which has significantly contributed to the improvement of the manuscript.
Sincerely,
Corlateanu A.
On behalf of all authors